# Livelihood, Market and State: What does A Political Economy Predicated on the 'Individual-in-Group-in-PLACE' Actually Look Like?

**Stephen Quilley * and Katharine Zywert**

School of Environment, Resources and Sustainability, University of Waterloo, Waterloo, ON N2L3G1, Canada
* Correspondence: squilley@uwaterloo.ca

**Abstract:** Ecological economics has relied too much on priorities and institutional conventions defined by the high energy/throughput era of social democracy. Future research should focus on the political economy of a survival unit (Elias) based upon Livelihood as counterbalance to both State and Market. Drawing on the work of Polanyi, Elias, Gellner and Ong, capitalist modernization is analyzed in terms of the emergence of a society of individuals and the replacement of the survival units of place-bound bound family and community by one in which the State acts in concert with the Market. The operation of welfare systems is shown to depend upon ongoing economic growth and a continual flow of fiscal resources. The politics of this survival unit depends upon high levels of mutual identification and an affective-cognitive 'we imaginary'. Increasing diversity, a political rejection of nationalism as a basis for politics and limits to economic growth, are likely to present an existential threat to the State–Market survival unit. A reversal of globalization, reconsolidation of the nation-state, a reduction in the scope of national and global markets and the expansion of informal processes of manufacture and distribution may provide a plausible basis for a hybrid Livelihood–Market–State survival unit. The politics of such a reorientation would straddle the existing left–right divide in disruptive and unsettling ways. Examples are given of pre-figurative forms of reciprocation and association that may be indicative of future arrangements.

**Keywords:** ecological economics; Karl Polanyi; Norbert Elias; Ernest Gellner; civic nationalism; survival unit; Walter Ong; Livelihood; reciprocity; conservatism; socialism; distributism

## 1. Introduction

In what follows, we argue that any effective agenda for ecological economics as a discipline, and the sustainability movement more generally, must attend to the potential of Livelihood to act as a disruptive 'third leg' within an ecological political economy. Socially and spatially mobile modern individuals depend, for the first time in human history, on a 'survival unit' based on an amalgam of both State and Market. The opposition between State and Market in contemporary left–right politics obscures the symbiotic relation between these institutional and ideological poles, both of which require economic growth to secure material and social wellbeing. The reassertion of Livelihood—i.e., the household, the informal/DIY economy, and the culture and rituals of reciprocation—is disruptive in that it both precludes many cherished policy commitments associated with social democracy, whilst opening up opportunities for political–economic transformation that have greater overlap with anarchist, paleo-conservative, religious and libertarian traditions. Building on ideas developed in a sequence of papers published over the last five years [1–8] and taking a cue from both Polanyi and the early Herman Daly, this paper will argue that Livelihood is a foundational domain—necessary to reign in both State and Market, and to create a 'three-legged' political economy for an alternative

modernity. Grounded in a triptych survival unit composed of Livelihood, State and Market, this alternative political economy has the potential to realize many of the ecological and social goals championed by ecological economics and the broader field of sustainability. Making this adjacent possible a reality, however, requires a more nuanced understanding of the complex tensions involved in rebalancing the domain of Livelihood with the interconnected domains of State–Market and cannot be accomplished without drawing insights from across the ideological landscape. This paper expands on our previous work by identifying the basis for a more fruitful discussion between small-c conservatives on the one hand and social-democratically minded greens on the other. Since any real paradigmatic, peaceable transformation toward a sustainable society would require bringing a critical mass of the population in high-income countries on board, and given the populist shift we are currently witnessing all over the world, this is a conversation that sustainability advocates cannot afford to avoid. For ecological economics more specifically, this conversation is necessary if the discipline aims to reconcile the dilemmas of political, economic and societal organization with global ecology and sustainability.

The a priori liberal-democratic commitment to an ontology of individualism is a necessary prerequisite—not just for the global consumer society, but also for the functioning of liberal democracy and law, as currently conceived. But this same structure of individualism is created by, and depends upon, an exclusive and monopolistic relationship between the Market and the State. Rather than antipodean opposites, the State and the Market are in fact mutually constitutive of each other. Only Livelihood offers the possibility of moving toward a low energy/material throughput version of modernity. But at the same time, the politics of Livelihood potentially destabilizes 'business as usual', left–right politics. It suggests common ground with some socially conservative and libertarian strands on the right, whilst requiring difficult conversations with many on the left, with regard to the ontology of individualism that underlies the unconstrained agenda of both human rights and consumerism.

What follows is not an empirical study, but rather, a theoretical analysis of the landscape that draws on historical sociology to trace the emergence of the 'society of individuals' and the Market–State nexus as a modern survival unit. The paper then attempts to discern the parameters of an adjacent possible ecological political economy. To illustrate how the domain of Livelihood can support an alternative modernity, we will draw on contemporary and historical practices for health and care that challenge the ontological bases of the society of individuals. Finally, we outline the theoretical touchstones of a doctoral project focusing on Livelihood as the basis for a radical political economy.

## 2. Elias and Polanyi on the Sociology of Modernization: The Great Transformation, the Counter Movement and the 'Society of Individuals'

From the mid-nineteenth century, sociologists struggled to get to grips with the expansive, dynamic and corrosive quality of capitalist modernization. Attention focused variously on: the conflictual dynamics of class society (Marx); individualization, rationalization, secularization and 'disenchantment' (Weber); 'anomie' and the problematic relation between new free floating individuals and society (Durkheim); the tension between the cohesive character of traditional rural and small town community and the dynamic instability of urban cosmopolitan society (Tönnies); the long term internalization of constraints on individual behavior and the growing complexity of market–state society (Elias); and the shift from status to contractual forms of integration (Maine). Building on Marx and Weber, one of the most illuminating accounts was provided by Karl Polanyi [9]. In The Great Transformation, Polanyi focused on the disembedding of the economy from social, cultural and religious structures of society. A society is 'modern,' he observed, when the domain of 'economy' is clearly understood by participants as a separate domain with its own laws, conventions and dynamics. This idea would have been quite literally unthinkable to Malinowski's Trobriand Islanders or to medieval peasants or theologians. Markets and market places are probably as old as humanity. But price-setting markets wherein connections are severed between supply and demand on the one hand and the conventions, cultural practices, ontological commitments and mythologies associated with

particular places, patterns of authority and kinship networks on the other hand—this was, for Polanyi, the disruptive feature of early-modern Europe.

Polanyi's study focuses on the effects of the enclosure movement in the English countryside. Nascent forms of capitalist agriculture saw the appropriation of common lands, the removal of peasants from the land and the unraveling of traditional forms of community and authority. Although unequal and hierarchical, the latter were cohesive and involved bonds of mutual obligation between landed aristocracy and peasants. Involuntary emancipation of the latter was experienced as brutal exposure to the vagaries of the emerging Market Society. In this first round of what Marx referred to as 'creative destruction', newly landless and footloose peasants became free wage laborers, driving a corollary process of factory development in the rapidly expanding cities of the north and midlands.

It was this combination of push and pull factors made possible by the disembedding of individuals from the close, cloying and sticky web of rural relations, that drove the emergence of the now ubiquitous relation between the individual and the State–Market. Initially, this unfolded only in reference to the market. Although Marx emphasized the generative possibilities of class consciousness associated with the shared social experience of the 'satanic mills' (William Blake's phrase) and the densely packed slums that surrounded them, in hind-sight, much more important was that workers interacted with even these emerging collectivist forms of association as individuals, reflecting the individual contractual wage relation (this was an aspect of what Maine referred to as the shift from status to contract).

However, as Polanyi showed, the nineteenth century liberal project of an economy and society of individuals, was utopian. If carried to its conclusion, it would have destroyed the ecological and social basis of society. To some extent it was this utopian belief in the power of market individualism, rather than malevolence, that led to the catastrophic response to the famine in Ireland. But with the specter of genocidal mob violence in the French revolution still raw, episodes such as the Peterloo massacre in Manchester, the unprecedented scale of Chartist mobilizations for the extension of democratic franchise and the growing strength of working-class trade unionism and syndicalism, it became very clear that market individualism would have to be complemented by some kind of social compact between the state and citizenry.

More generally, the triumphalist liberal ideology was complacent and therefore underestimated the role of the state in 'instituting' the market society [10]—through the creation of a monopolistic currency and central bank, the establishment of a monopoly of violence and the peaceful and predictable context for trade, the use of police and legal infrastructure to ensure that contracts were honored, the innovation of copyright law and legal fictions such as the joint stock company, and ultimately by providing a new social compact based on physical and economic security for citizens.

In the end, it was this social compact that averted social and political catastrophe and collapse in countries such as Britain and Scandinavia (and failures in this regard that saw totalitarian reaction in Russia, Italy and Germany). From the middle of the nineteenth century, what Polanyi [9] referred to as a 'countervailing movement for societal protection' engendered what would eventually become the protective carapace of the Keynesian Welfare State. This included interventionist macro-economic policy to secure full employment, universal healthcare systems, social insurance, social housing projects, the regulation of the workplace and the working day, minimum wage legislation, laws against discrimination on the basis of sex, race, religion and disability, legal aid, and a raft of other institutional extensions of what Bourdieu [11] called the 'left hand of the state'.

## 3. The State-Market as a Survival Unit

From a sociological perspective, what emerged over this period of centuries was what Elias [12] called 'the society of individuals.' The key feature of this new kind of society is the nature of the 'survival unit' (see Figure 1, Box 1). In traditional society, individuals depend on each other—on their immediate kith, kin and community relationships that are daily, weekly, annually and generationally reciprocated in the context of particular places, in what is for the most part, a face-to-face society. By contrast, the defining feature of modernity is that individuals are cut loose. The assumption of market

society is one of a ubiquitous spatial and social mobility. In contrast to pre-modern peasants, modern individuals can enter freely into contracts of employment and investment. They can climb the social ladder. They can marry or unmarry at will. They can move without restriction. But of course, with this freedom came great economic insecurity. In traditional rural society, the constraints on mobility came with obligations on the part of landowners, extended family members and neighbors to support and defend individuals within their network. In the society of individuals, the survival unit in the first instance centered on a person's relation to the labor market. As Tory Minister Norman Tebbit said to redundant miners in Thatcherite Britain, "get on your bike"—a sentiment echoed rather more poetically by Janis Joplin's lyric "freedom's just another word for nothing left to lose."

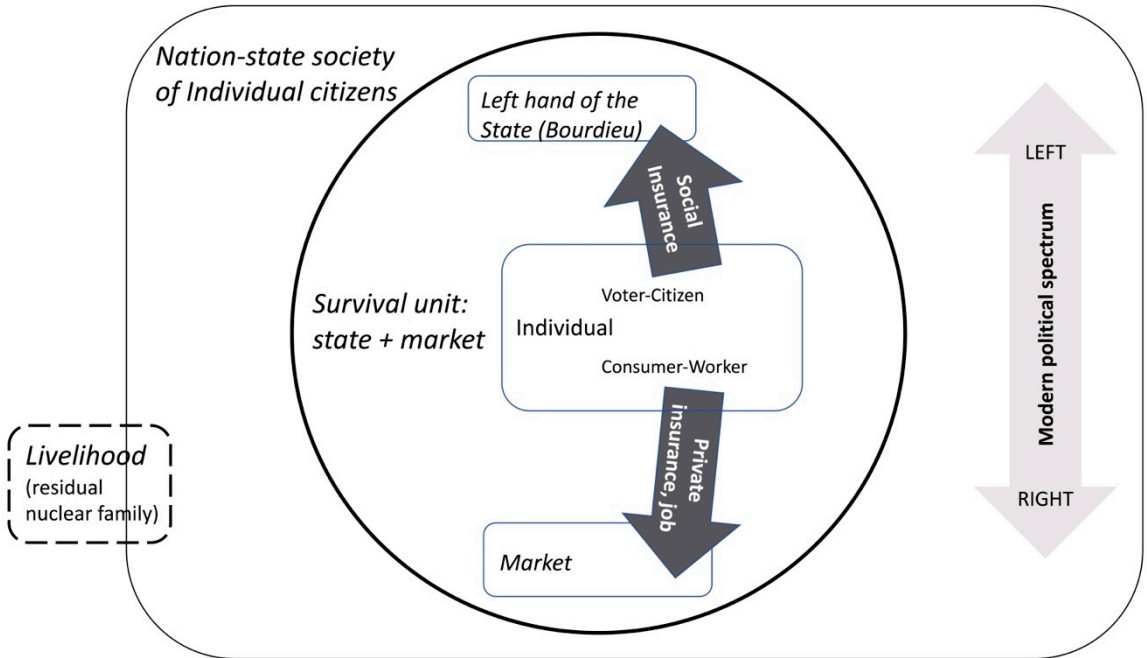

**Figure 1.** State–Market: The modern society of individuals.

In the end, it was the precarious nature of this survival unit and the political consequences of such insecurity that saw the unfolding of Polanyi's 'countervailing movement for societal protection'. In the century between the publication of Engel's The Condition of the English Working Class in 1844 and the 1942 Beveridge Report (preparatory to the establishment of the post-war welfare state), the gradual elaboration of the state apparatus deemed necessary to stabilize economy and society amounted to the emergence of a new kind of survival unit, namely the State–Market. In modern western-type societies, the physical and economic protection of individual citizens is achieved through a combination of market access (employment, shares, investment opportunities, pensions, private insurance) and state sponsored social insurance, welfare benefits and entitlements. In theory (though not always in practice), no citizens should ever fall through the net.

**Box 1.** Summary, State-Market Survival Unit.

---

In modern gesellschaftlich societies, the 'I' is partially disembedded from the 'We'—and to an even greater extent in the self-perception of most modern individuals as well as the epistemological and foundational assumptions of modern disciplines such as economics (Homo economicus) and philosophy (Cogito ergo sum), and institutions such as the legal system or democracy. Individuals experience unprecedented degrees of social and spatial mobility. Disembedded price-setting markets predominate over convention-bound market places. An intrusive polyvalent state intervenes overtly and behind the scenes in all areas of social and economic life. The we-imaginary of family and place is supplemented and displaced by the quasi-familial 'imagined community' of the nation (Anderson)—with narratives organized around the mythological heroic tropes of shared-language and national history (Gellner). For citizens tied to this new social compact, the survival unit centres on individual contracts with State and Market. Mobile individuals experience a more autonomous, interiorized and often vulnerable sense of sense of self. The erosion of common pool resources goes hand in hand with the erosion of face-to-face relations of mutual obligation and interdependency. The significance of personalized interaction gives way to that of intensified, abstract, opaque relations of interdependency over extended space.

---

## 4. Sociogenesis and Psychogenesis: The Novelty of 'the Individual' as Both a Unit of Analysis and a Neuro-Psychological Personality Type

One under-appreciated aspect of the society of individuals, is that it has engendered a novel and historically unprecedented personality type. It has been well-established over a long period of time, by work associated with the 'culture and personality' school in Anthropology, that different kinds of society engender very different forms of psychology and outlook. Certainly, the extremes of social constructionism have been discredited rather comprehensively [13], not least with Freeman's demolition of Margaret Mead [14,15]. The mistake of social anthropologists, and sadly the majority of sociologists, is to disregard entirely any possibility of an evolved human nature. Nevertheless, evolutionary psychology notwithstanding, there is an astonishing plasticity in behavior and outlook consequent on language-culture (albeit within limits). Elias [16] attended to this plasticity within limits with the concept of 'second nature'—the process whereby external social pressure, culturally specific cognitive framing and endless repetition in the context of early socialization (the 'ontogeny' of individuals) effectively hardwire culturally specific forms of behavior and cognition that become so automatic as to appear (if it 'appears' at all) to societal insiders as 'natural'. Most of the time, such naturalized patterns remain below the level of conscious reflection.

In *On the Process of Civilization* [16], Elias demonstrated how, over many centuries, the increasing complexity of European state–market societies had seen the loosening of local, social and physical constraints on individual mobility (Polanyi's process of 'disembedding') accompanied by, first of all, greater external constraints on behavior exercised by increasingly powerful and monopolistic states, but subsequently by the internalization of self-constraints on the part of individual citizens. In short: as the 'I/We balance' shifted toward the former; and the intricate web of ascriptive local and familial ties loosened to allow individuals to move, make choices and for the first time see and understand themselves as individuals; so, these individuals were now constrained by formal and contractual mechanisms and laws of the 'central power'—and overtime, from within, by the elaboration of a more extensive and vigilant superego mechanism. In this way, the sociogenesis of the State and the Market society depended upon a corollary process of psychogenesis that was characterized by:

- More extensive individuation
- A more elaborate and intrusive super-ego, policing behavior and restraining impulses
- Less behavioral volatility
- Greater capacity and necessity for 'detour behavior' or the deferral of gratification
- Greater self-awareness
- Interiorization of mind and a heightened sense of self as a discrete entity
- Greater understanding of the meaning and course of an individual life as separate from and not reducible to the life-chances and prospects of an immediate survival group

Elias's elaboration of Weber's account of individualization in modern society is sophisticated and complements very well the history and political economy of disembedding given by Polanyi. His thesis is also remarkably congruent with another body of work associated with Walter Ong. In Orality and Literacy [17], using examples from Homeric era Greece to oral poetry in twentieth century Nigeria, Ong explores the impact of literacy on psychology and culture. His findings are both astonishing and pertinent.

Ong's thesis is essentially that reading and writing transformed human consciousness, changing how human beings think and behave—to such a great extent, that it is almost impossible for a person of a literate culture to look at or hear the world through the eyes and ears of someone from a primary oral culture. Literate cultures have been around for only several thousand years—a mere blip in human development. But modern societies based on universal literacy began to emerge only in Early Modern Europe (see below). The critical feature of writing, Ong argued, was that this medium or mental expression began to transform the mind—'the most momentous of all human technological inventions . . . Because it moves speech from the oral–aural to [the] sensory world . . . of vision, it transforms speech and thought as well.' [17].

In a primary oral culture, thought and speech tend to be 'additive' rather than 'subordinative,' mitigating against complex abstract, analytical and hierarchical descriptions and arguments which are impossibly difficult to sustain in the ephemeral ether of the oral-aural universe. Speech is dominated by redundant and copious forms of expression. It tends toward conservatism, being dominated by the twin imperatives of the immediate situation and bodies of thought and expression inherited from previous generations. For the same reason it is characteristically close to the quotidian context of the everyday life world and mitigates against abstract thought experiments or counterfactuals. And oral expression tends toward what Goody and Watt [18] call 'homeostasis'—a present-centred equilibrium that is maintained by off-loading and dispensing with memories no longer relevant. Uninterested in definitions [19], the meaning of words emerges always and only from the context of habitual use—an expressive niche that includes other words but also gestures, vocal inflections, facial expression, and the cultural and cosmological setting.

In an oral society, language is ephemeral. Speech sounds dissipate immediately. And as a result of the premium on memorization, there was an enormous imperative to think and express memorable thoughts involving repetition, amplification, recurring tropes, standard mnemonic patterns, looping, dense associations—patterns shaped for oral recurrence and aural transfer to other people. One consequence of literacy is to downgrade the utility and cultural value of old people, 'repeaters of the past', in favor of younger people who are potential 'discoverers of something new.'

In ontological terms, the immediacy of the aural–oral world construes the cosmos as an ongoing event with man at its centre, 'the navel of the world.' In contrast, the ocular vision of printed word and later maps predisposes literate people to think of the world as something 'out there'—an external surface ready to be explored and transformed, and the past and future states of which can be tracked from without.

As solitary activities, writing and reading engender introspection and foster that internal 'voice' that moderns take for granted. Writing allows thought processes to be documented, organized and made linear by individuals working alone. By contrast, oral expressions are public and shared. Thought is necessarily a dialogical, iterative, collaborative and shared process. Similarly, literate cultures internalize emotions; oral cultures externalize them.

In this context, it is easy to understand the emergence of modern literature and the novel in terms of the interiorization of human subjectivity made possible by writing. With a definite beginning, middle, and end, writing construes a story as authored, self-contained, discrete and defined by closure. In contrast, Homeric era oral poetic epics had no plots as such. And if modern literature provides an index of the disembedded, sovereign individual subject, the scientific worldview is even more a function of literacy and writing. Rather than trading in verifiable 'facts', the means of orientation associated with inter-generational memory centres on parables, proverbs and poetic associations.

## 5. Literacy, Modernization and the State

We are now in a position to reflect on the relationship between the psychogenesis of modern individuals and the nature of the State–Market survival unit. As indicated above, traditional survival units were overwhelmingly local, community and kin-based (Figure 2; Box 2).

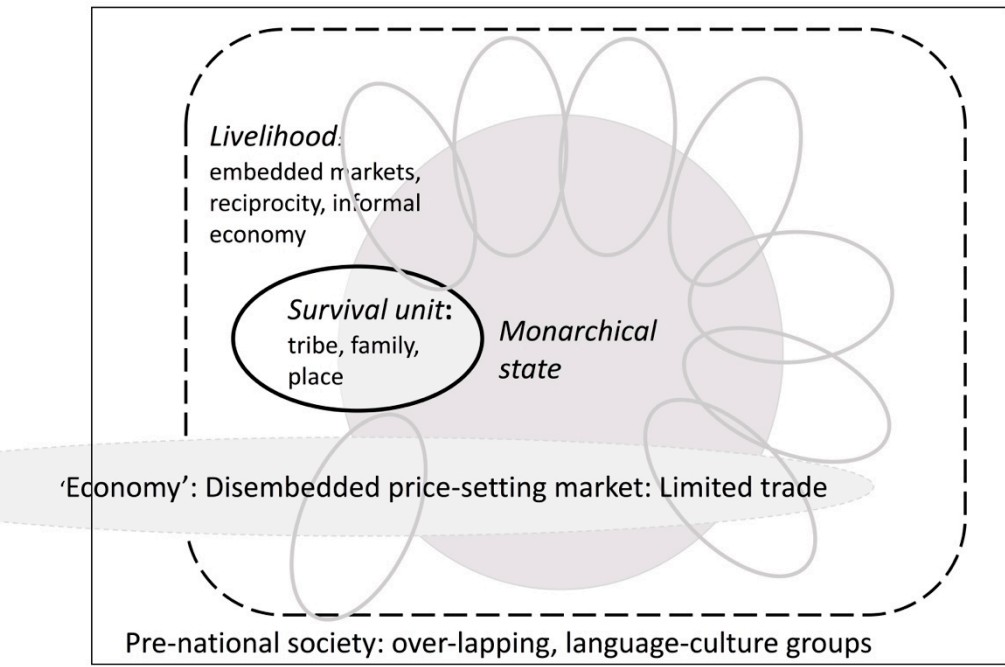

**Figure 2.** Traditional Livelihood Society.

**Box 2.** Summary, Traditional Livelihood Survival Unit.

In traditional 'gemeinschaftlich' societal form, the 'I' is firmly embedded in the 'We' and the survival unit is local, familial, tribal and place bound. Market activity is for the most part embedded in a tissue of cultural, ritual, ontological and social relationships (Polanyi). There is widespread use and dependence on the 'commons', along with ubiquitous patterns of reciprocity and gifting. The central state is distant and minimally intrusive in the day to day life-world and economy.

'Societies' in the modern sense did not exist. There were, of course, often monarchical states and empires that controlled swathes of territories. However, the borders of such entities were often fuzzy and ill-defined. Gellner [20] contrasts the nature of the linguistic high culture (language, religion, art, literature) that characterized the military, aristocratic and theological elites of such societies, with the parochial and illiterate localism of the peasant communities upon which they depended. Such high cultures often straddled borders as with Catholicism, intermarrying dynasties, fashions in luxury goods or the use of a 'lingua franca' such as Latin or later French as opposed to one of myriad local vernaculars and dialects which shaded into each other. Thus, typically in such societies, there was a sharp cultural and often ethnic differentiation between a horizontally stratified elite of warriors, nobles, priests, merchants and administrators on the one hand, and vertically stratified peasant producers on the other. The 'high culture' of the latter engendered a broad cosmopolitanism and abstract cognitive framing of human social life and cosmology—a nascent form of 'context-free communication'. Peasants, in contrast, were defined by an intense localism and place-specificity. Such communities are very unlikely to understand their own idiosyncratic cultures as being linked by a political principle to other equally particularistic and incommensurable local cultures. Such a proposition would have appeared almost absurd or unthinkable to participants. As Gellner puts it, 'The self-enclosed community tends to be communicated in terms whose meaning can only be identified in context, in contrast to the

relatively context free scholasticism of the scribes ... The village patois (or shorthand or 'restricted code') has no normative or political pretensions' [20].

As Geller argues, the defining feature of modern nation states was the imposition of a single high culture over a defined territory. More than anything else, this was achieved through the vehicle of literacy. The reproduction of individuals has always been achieved in one (or a mix) of two ways. In pre-modern agrarian societies, education was accomplished through one to one, intra-community means. Such embedded processes of acculturation depended on self-reproducing and for the most part self-sufficient social units, in which the process of socialization and maturation was simply an aspect of the unfolding of the whole life of the community. By contrast, for a tiny elite, the necessity of mastering an increasingly sophisticated high culture engendered new forms of scholastic and formal 'exo-education.' In this 'centralized method,' local acculturation is supplemented or even completely replaced by a dedicated education or training function—an institution that is 'distinct from the local community and which takes over the preparation of the young human beings in question, and eventually hands them back to the wider society to fulfil their roles in it when the process of training is complete' [20]. In the case of specialized exo-training, skills are brought in from outside. He gives as an example the 'Devshirme' system of the Ottoman Empire, whereby young boys secured as tax obligation from conquered populations or purchased as slaves, were systematically trained for war and administration and ideally wholly weaned and separated from their families and communities of origin. Similarly, the boarding schools and residential universities servicing the British upper class from the eighteenth century can be understood as very clear examples of exo-training.

In his account, Gellner highlights the driving imperative of 'context-free' communication as a prerequisite for the processes of rationalization and marketization. In ancestral, face-to-face societies, communication was highly specific to time, place and social context. Tone, gesture, personality and situation were indivisible from content. Utterances and messages were often polyvalent and overlaid with meaning. Only the scribes of a society's high culture—the lawyers, accountants, administrators, theologians, and ritual specialists—might use something approximating to an explicit, rule-bound and precise, context-free form of communication. For the great majority, communication remained intimate, implicit and saturated with diffuse meaning. In many contexts (such as the exchange of gifts), too much precision would have appeared pedantic, offensive and even unintelligible. For countless millennia, language operated in this intimate, face-to-face and contextual manner. It was not until the onset of capitalist modernization that the powerful but also corrosive potential of context-free communication was fully realized. In conditions of capitalist modernization, the expansion of complex formal trading and market arrangements linking the daily activities of millions of citizen-workers; and given the imperatives of nation-state formation, effective communication required language and expression that minimized colloquial, ill-defined, contextual meanings to allow the precise coordination of activities over ever greater distances. The rationalization of language through literacy, dictionaries, standardized spelling and the codification of grammatical rules, in this sense, went hand in hand with the standardization of national weights and measures, clock time, legal codes and contractual conventions.

## 6. The Importance of Civic Nationalism and 'National Society' as the Container for the Social and Spatial Mobility of Individuals with Rights

To recap, from Polanyi [9], we have an account of disembedding of the economy involving the dispossession of peasants and the forcible creation of a society of spatially and socially mobile individuals. From Gellner [20], it becomes clear that literacy through exo-education becomes the most significant vehicle for the imposition of a unitary 'high culture'—both in allowing the emerging market economy to function efficiently and by smoothing the barriers to capital and labor mobility. From Ong [17], it is equally clear that mass literacy has had a transformative impact on the consciousness and personality structure of people in technically advanced societies. And finally, from both Polanyi and Elias [12,16], we have a persuasive account of the way in which the resulting society of individuals

is predicated upon, and requires, a very different kind of survival unit (Figure 1; Box 1). Specifically, having sundered the communitarian bonds of reciprocal care and obligation, modern individuals became quickly both dependent on and vulnerable to the capricious cycle of market forces. Through decades of institutional innovation, political agitation and elite compromise, modern societies have become defined by a new survival unit—that of the State–Market (see Figure 1).

Finally, it is important to dwell for a moment longer on what Elias called the I/We balance. In the society of individuals, this balance tilts decisively toward the former pole, and against the ascriptive capacity of small-scale, place-bound structures of mutual-identification to enroll, cajole and sometimes coerce individual compliance with the group. This process is synonymous with a certain idea of freedom that has, for centuries, animated rural people to leave their villages for the bright lights of the city. It is a freedom that Marshall Berman [21] notes often has tragic consequences, particularly for women (as with Faust's first love Gretchen). And yet modernity is defined by this irrepressible 'shout from the street'—the combination of push and pull factors that force and entice modern people to try, often against the odds, to make themselves the subjects rather than the objects of broader historical processes. But against this crescendo of individual self-actualization, the 'we' does not, and cannot, disappear. Survival groups are by definition social units in which individuals forego the 'bean-counting' rationality of trade and barter in favor of a kind of pooled sovereignty predicated on everyone 'being in the same boat'. If the survival units associated with small place-bound communities no longer deliver security guarantees and are less able to command allegiance and conformity—this is only because they have been replaced by the higher-level survival unit of the State–Market (see Figure 1).

A woman in twenty first century North America, Japan or Europe can study for a degree, take a high-paid job and choose to have a baby by artificial insemination. By historical standards her chances of violent rape or assault are low enough that she can afford to dispense with a husband, extended family or chaperone—confident in the expectation that, for the most part, the streets are safe and she can manage her own affairs. As a survival unit, the modern nation-state–market delivers physical safety—but also childcare, healthcare, education, a pension and somewhere to live (all delivered by some combination of public or private 'contracts' consequent on employment or citizenship status). However, just like their premodern tribal and clan antecedents [22], the functioning of this new state–market survival unit depends on two related processes of legitimation.

First of all, the system has to work and deliver basic security. If the state fails to deliver security or welfare or the market full employment, then very quickly class and ethnic tensions that are often just below the surface of liberal consensus politics, come to the surface. This is what Habermas had in mind when he referred to the permanent possibility of a legitimation crisis [23]. The role of state-sponsored mass consumption in the legitimation of capitalist societies was laid bare, with almost comedic clarity, by Edward Bernays in his pamphlet on the 'engineering of consent' [24] and 'Manipulating public opinion. The how and the why' [25,26] (see [26] for a brilliant exposition).

The second form of legitimation relates to the process of taxation and redistribution—both in terms of public infrastructures and also cash and in-kind welfare payments to individuals. Within a family or a clan, reciprocity is the norm. And for most of human history, reciprocity and gifting are the normal mechanisms for the allocation and distribution of goods in the subsistence economy. The transfers necessary to fund higher-order forms of complexity associated with agrarian empires or Kingdoms were made possible by a combination of violent appropriation ('tax' is really only legitimate robbery) and often the cloak of religious authority (as with divine kingship). With modern societies there arises a new problem, namely the need to legitimate state appropriations and expenditures on a scale unprecedented in human history, but in the context of a nascent process of democratization. Elias [27] has discussed the tendency for complex societies to inch toward what he called a 'functional democratization' consequent upon the increasing interdependence between individuals and groups, the more opaque patterns of cause and effect, and the possible severity of unintended consequences, which together force more powerful groups and individuals to take gradually more account of less powerful ones. It is certainly true that from the eighteenth century, nation-states increasingly seek to

establish a direct and personal relation between the individual citizen and the state that is predicated and framed in terms of a symbolic consanguinity. Nations are construed as families and fellow citizens as brothers and sisters. A sister or brother co-patriot can no longer demand care and material aid from a relative stranger based on a face-to-face relationship and courtesy of a clan-relationship. But they can access the same from the state on the basis of a symbolic familial relationships and affiliation with the exclusive membership club that is the nation state. And the state can, by the same token, command legitimate wealth transfers from all citizens as a function of that same club membership. In highly functioning societies such as Canada, Scandinavia, the UK or Germany, this affective loyalty is so comprehensive and psychologically ascriptive that tax avoidance is very low, compliance is high, and public services are comprehensive and, by any historical standard, lavish beyond compare.

Such legitimacy in the eyes of a well-educated public is achieved by virtue of what Benedict Anderson [28] called 'imagined community'. A critical aspect of nation-state formation involves the elaboration of an extensive repertoire of origin myths, stories, a shared history, common language, a cannon of national art and music, national costume, national sports—all designed to foster and sustain a primal framework of mutual identification. As Hulme pointed out, the propensity for projecting community is an evolved aspect of human sociality. The boundaries of any particular community depend upon cultural and political processes [29]. Nation-state states must try hard to effect an internal homogeneity, strong semi-permeable borders and a sharp contrast with neighboring nation-states. It is for this reason as well as the functional requirements of rationalized economies, that the processes of exo-education described by Gellner, are so often so brutally attentive to the elimination of 'unofficial' language-cultures and achieving a monopoly-status for which ever high culture has come to define the trajectory of nationalization.

## 7. Social Democracy at the Cliff Edge: Individualism, Diversity and Left Identity Politics

So, this was the nature of the social compact for most of the period since the second world war. A society of socially and spatially mobile individuals whose freedom depended on the effective functioning of a State–Market survival unit: state institutions would provide material and economic security to supplement the opportunities provided by Market Society, whilst economic growth provided a steady flow of fiscal resources to support state expenditures. This structure of individualism was in other words supported by two forms of we-identification. The national we-imaginary has been discussed above. The second relates to the collectivism of class politics associated with organized labor and the array of social democratic parties that often dominated the electoral landscape in many European countries.

In practice, this social compact involved a balance between free-wheeling individualism on the one hand, and these two ascriptive we-imaginaries of nationalism and class collectivism, on the other. But from the 1980s on, with processes of globalization and the ascendency of neo-liberal orthodoxies promulgated in particular by Margaret Thatcher and Ronald Reagan, both of these we-imaginaries began to lose traction. As social democratic parties (notably the Democrats under Clinton and the British Labour Party under Blair) began to abandon the regulatory shibboleths of Keynesian economics and to adopt supply-side, market friendly policies in-tune with the 'realities' of globalization, so the unit of 'national society' and progressive nationalism began to give way in the progressive mindset to a kind of cultural globalism, soft internationalism and cosmopolitan rhetoric about multiculturalism and the value of diversity. Since 2008, the populist backlash has at least partly been fueled by right-wing parties occupying the abandoned space in the discursive landscape, i.e., of national society as a fundamental survival unit and prerequisite for the material security of working-class people.

At the same time, and somewhat ironically, even in the period of post-war 'welfare collectivism', liberal and social democratic politics did not abandon the atomistic vision of a 'society of individuals'. Social democracy actively participated in the social policy innovations that curtailed the bottom-up, self-organized power of parents, communities and churches in favor of universal provisions organized through the state [1]. More recently, contemporary visions of social justice pursue this, hybrid

Hayekian–Rousseau-ian vision of the sovereignty of individuals to its logical conclusion. But (clearly both contra Hayek and Rousseau), they pursue their vision through the state and against the residual gemeinschaftlich mores and associations of traditional society. Activists and bureaucrats alike, actively (sometimes intentionally) push for the unravelling of all the normative constraints on individual action associated with traditional society, whilst imposing a catalogue of new norms backed by the state (e.g., in the sphere of sex education, hate speech, gender pronouns etc). Taken-for-granted policy commitments tacitly promote a rational-instrumental, permissive, transactional and individualist understanding of marriage, the family, sex, work and community. The shared assumptions of the 'business as usual' worldview emphasizes abstract and universal rights, whilst consistently de-valuing local, concrete obligations as the incarnation of lived patterns of reciprocation between individuals embedded in particular sets of relationships.

With regard to the taken-for-granted arena of progressive social democratic politics, these developments do not bode well. Firstly, in abandoning the language of national society and civic nationalism in favor of a kind of cultural globalism, all over the world the left has become a fairly explicit fellow-traveler of neoliberalism and market conservatism. Cultural cosmopolitanism and increasingly 'woke' forms of genuflection to identity politics provide a fig-leaf barely concealing an active acquiescence to globalization and a vision of open borders and the free movement of both capital and labor. Tony Blair, the Clintons, the EU establishment, Justin Trudeau—paragons of global progressivism have, for decades, been quick to defend the interests of refugees and multiculturalism whilst denigrating any form of progressive nationalism and actively seeking to orchestrate trade deals that further consolidate corporate power whilst undermining the capacities of the state to regulate the market.

Over time this commitment to cultural globalism is undermining the functioning and legitimacy of the national social compact as a survival unit. The 'counter-movement for societal protection' which led to these national social compacts didn't emerge from nowhere. It was a response to the very real likelihood of class conflict, revolutionary disorder and violence that were a regular feature of industrializing societies from 1789 right through to 1945. Before the war, existential class violence was endemic in even the most stable nations, such as Norway and Sweden.

The irony is that the left-identitarians put forward a vision of individual freedom that is unprecedented even by twentieth century standards (including the freedom to change gender and abolish the concept of 'assigned sex')—but a vision that depends almost entirely on the public health and rights-based legal infrastructures developed by civic national-societies in the twentieth century. These depend in turn on an unfailing flow of fiscal resources from a growing economy and are legitimated by a solid and cohesive national 'we-imaginary'. But just as class collectivism was subdued and impoverished by the process of deindustrialization from the 1980s, so the ideological assault on civic nationalism and the increasingly dominant strain of racialized identity politics coming from university campuses (with the woke lexicon of privilege talk, intersectionality, victim hierarchy and an always under-specified 'decolonization'), combined with large scale migration in European countries, is now undermining this second pivotal source of class-based legitimation for nationally-based fiscal transfers and expenditures. Social democracy has always depended on the exclusive form of solidarity that can be generated on the basis of shared citizenship—itself an exclusive membership club. The more diverse a society is, the more difficult it is to enroll different groups into a shared legitimating we-imaginary. It is not accidental that America, bearing the original sin of genocide and slavery and a propensity for hyphenated qualified American identities, has excelled in the politics of expression but at the cost of a much inferior welfare state. And likewise, the universal and generous cradle-to-grave welfare systems of Scandinavian countries are to a large measure a consequence of their small size, ethnic homogeneity and distinctive language-cultures that facilitated the orchestration of effective and monopolistic national we-imaginary. It seems likely that with the enormous influx of refugees since 2015, Sweden and Germany in particular may struggle to sustain the cohesive national we-imaginary that has upheld the welfare system since the war. Other things being equal, such an increase in

ethnic and religious diversity would necessitate more intrusive, and even coercive, state interventions to secure the necessary level of mutual identification—this or a rapid erosion of the social-welfare consensus and a politics dominated by ethnic sectarianism and tax revolts.

## 8. Social Democracy at the Cliff Edge: Ecological Limits

Unfortunately, other things are not equal. From an ecological–economic perspective, the world is moving rapidly into an era of biophysical limits to growth [3,30–35]. Without laboring the point, it is very obvious that the State–Market survival unit—the basket into which the traditional social democratic left, classic liberals, Christian democrats and more recently, the left-identitarians, have put all their eggs—simply cannot survive in its present form. For this reason alone it is curious that Green Parties talk up the most catastrophic time-lines for climate change and crashing biodiversity, whilst, at the same time, advocating with a blind passion the extension of all sorts of public infrastructures from free university education, state child care, health provisions and in the UK an unwavering commitment to the European Union—despite the fact that for political reasons, the EU must be more committed to growth at all costs than any of its component nation-states (without the capacity to redistribute fiscal surpluses between member states, the EU would run into a terminal and existential legitimation crisis within a matter of months).

We have explored the tension between the ecological politics of growth and the welfare state extensively elsewhere [1–5]. Here, it is sufficient to reiterate that if ecological economists are serious about limits to growth, they must necessarily be willing to rethink the nature of the welfare state and more generally the kind of survival unit that is conceivable and possible on the one hand and perhaps more (or less) palatable on the other. In what follows:

(i.)　We outline an alternative survival unit that envisages the re-emergence of the domain of Livelihood as a balance to contracting domains of Market and State.

(ii.)　Identify novel political and ideological formations that may make such an eventuality plausible.

(iii.)　Detail some examples of pre-figurative experiments that may be indicative of future developments.

## 9. Livelihood-State–Market: A Lower Overhead Survival Unit

In the course of modernization, the emergence of a society of socially and spatially mobile individuals is very difficult to reconcile with tribal, familial or place-bound forms of organization. This is true for all sorts of reasons including:

(i.)　Any exposure to literacy undermines and largely eliminates the forms of consciousness associated with oral cultures [17,19].

(ii.)　Mobile individuals engaging contractually with price-setting labour markets tend to be mobile and to disengage from the viscous, sticky and time-consuming work of maintaining relations of reciprocation between groups and generations in particular places. An exo-education is a 'ticket to ride' rather than an invitation to stick around.

(iii.)　Modern polities are legitimated through claims to universalism in relation to law, entitlements and authority. Liberal democratic legal systems must, by definition, construe people as individual citizens. Tribal authority and obligations relate to groups and do not countenance any necessary solidarity between clans or tribes. The two systems are almost impossible to reconcile without inducing a structure of permanent reliance on the state (as with First Nations in Canada). At the same time, arrangements which allow differential outcomes or treatment depending on sub-group membership contravene the principle of individual equality that is a foundation for any concept of national citizenship. Any significant group-based differential undermines the legitimating function of the 'imagined community.'

Nevertheless, it is possible to envisage a modern state–market society operating on the basis of a radical principle of subsidiarity allowing all manner of hybrid arrangements to develop. A nation

state may be able to sustain a weaker, less-developed national we-imaginary, if at the same time it is having to sustain a significantly lower overhead of fiscal transfers between the formal economy and citizens. But of course, any such contraction in state services and infrastructures could severely undermine social cohesion and political stability—unless low-overhead systems associated with lower financial costs and much reduced ecological footprint, can be found to replace them. This, in effect, begs questions about a low-overhead survival unit.

During the early modern period when traditional structures and safety-nets were falling away, but before the decisive extension of Bourdieu's 'left hand of the state' [36], individuals and communities experimented with all kinds of institutional and social innovations directed at achieving some kind of material and physical security in a challenging, mobile and precarious world dominated by unpredictable market forces. Such innovations included guilds, friendly societies, insurance clubs and so on (see 1 for a discussion).

Over the course of the twentieth century, the two-dimensional ebb and flow of left–right politics has acted to obscure the systematic dismantling and exclusion of forms of association and mutualism that I have characterized (after Polanyi) as Livelihood—i.e., the household, the informal/DIY economy, and the culture and rituals of reciprocation. In technical terms, the State and the Market, as has been elaborated above, require and constitute each other. Rhetorical opposition aside, the Left just as much as the Right has been committed to the rendering of people as civic individuals. Certainly, socialist and social democratic politics have relied on the momentum achieved by combinations of such individuals exercising their potential veto over the labor process, to extract concessions in the form of state regulation and public infrastructures. However, the we-imaginary at play was not a bottom-up expression of mutualism and reciprocation, but was characterized by institutional forms of collectivism which involved the aggregation of (individual) citizen-power. Such organizations were built on the Enlightenment principles of individual rationality, choice and suffrage. Collective power involved the aggregation of these abstracted individual expressions of political will, rather than the affective-emotional, emergent group-consciousness characteristic of reciprocating, face-to-face communities.

Thought experiment: Figure 3 summarizes the parameters of a hypothetical triptych Livelihood–State–Market survival unit (also see Box 3). This scenario doesn't involve going back to some pre-modern arrangements. Rather it is predicated on hybrid political and institutional forms that although possible, have been rendered invisible by the priorities and assumptions of mainstream left and right ideological discourses. In systems language we are attempting to describe an 'adjacent possible'—a part of the landscape of possibility that is feasible and reachable, but for whatever reason has been hitherto culturally, cognitively or politically unavailable.

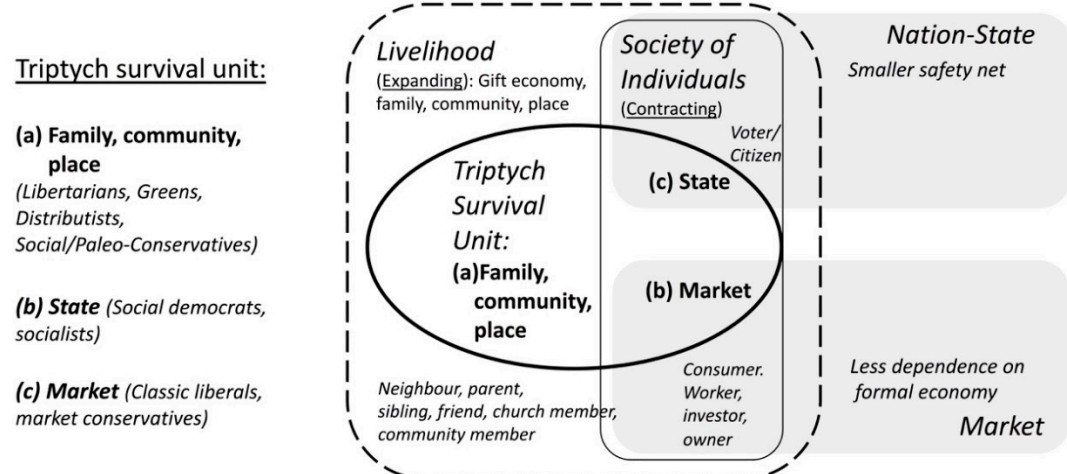

**Figure 3.** The Triptych Survival Unit: Livelihood–State–Market.

Driven by an intentional policy of degrowth or more likely a sharp disruption to the mainstream economic cycle, this scenario involves a reduction in the scale and scope of state activities consequent on a contracting economy and a large reduction in the tax base and fiscal transfers. At the same time, technical developments relating to micro-fabrication (synthetic biology, 3D printing) and the sharing of intellectual property (design, specs, process instructions) via the Internet, are beginning to make conceivable a wholesale informalization of whole swathes of production activity dominated by corporations for the last century, i.e., repair, modification and innovation of new products in domestic or community settings for local use or sale in the context of (more/or less embedded) market-places.

**Box 3.** Summary, Livelihood-State-Market Survival Unit.

A low/no growth society would see the re-emergence of the sphere of Livelihood in tandem with smaller and less intrusive State and Market systems. Construed as a hybrid of Gesellschaftlich and Gemeinschaftlich elements, relative to the early 21st century the I/We balance would shift toward the latter. The re-emergence of gifting and reciprocal exchange would facilitate greater dependence on an expanding web of informal and embedded markets. Nation-states would almost certainly remain the principle form of macro-organization and may even strengthen with a reversal of globalization, but in tandem with expanding subsidiary forms of mutual aid and identity at local and neighbourhood levels. The relation between the state, regions and community would be defined to a much greater extent by subsidiarity, allowing greater autonomy and self-management but also greater potential for inequalities between regions and communities. A central political-economic problem would be the re-emergence of common pool resources and legal/cultural mechanisms for managing them. Informalization of parts of the economy would be facilitated by emerging high-tech micro production tools such as 3D printing that make possible the manufacture and repair of sophisticated equipment bypassing corporate hierarchies in the formal economy. Reduced tax returns from a shrinking formal economy will lead quickly to a reduction in fiscal transfers and the simplification and trimming down of the welfare state. The politics of this transition are likely to be disruptive and even chaotic, creating space for novel and unsettling political coalitions and ideologies.

There is a significant but unknown potential in these emerging technologies to radically cut the 'transformity cost' of complexity. Transformity here refers to Odum's [37] analysis of the embodied energy cost that accumulates along all the myriad distributed production processes that are required to produce a given artefact. Thus, to produce a pint of milk from a cow, a subsistence farmer has to sustain a cow on a given acreage of grass and milk it. A fraction of the solar energy hitting the ground is captured by the grass; a fraction of the grass is eaten by the cow; a fraction of that energy is turned into milk. At each stage, energy is dissipated as heat. Odum's energy accounting calculates the embodied energy of an artefact or process in terms of standardized solar-energy units (SolarEMJoules). Apply the same accounting framework to a pint of milk bought from the supermarket and the transformity cost can be seen to escalate almost exponentially, for now the web of antecedent processes involved includes trucks to transport the milk, a factory to process it, trained truck drivers and factory operatives, colleges and universities to train these operatives—and so on.

Technological complexity is ecologically and metabolically expensive precisely because it escalates these embedded costs so inexorably. It would be almost impossible to track and account for every energy transformation and production activity in the economic web that is necessary to produce even a humble BIC biro. The significance of high-tech micro-fabrication rests on its potential (or not) to strip out swathes of these ancillary costs. It is easy to imagine keeping and milking a goat at home, but harder for most people to envisage breeding and killing a cow for meat, let alone repairing or even manufacturing a vacuum cleaner or a mobile phone—but, if not a Star Trek replicator, it is at least conceivable to start thinking in such terms. Synthetic biology and 3D printing may, before too long, allow for lab-scale meat synthesis or the production of printed circuit boards and silicon chips in domestic environments.

In an extended discussion of this kind of scenario, Carson [38] describes the potential for a low-overhead homebrew industrial revolution in which individuals, families and groups would be allowed, for the first time in a century, to use household assets (garages, kitchens, cookers, land) as a veritable capital and 'means of production' for a resurgence of 'household economy'

(the oikos). Drawing on Lewis Mumford [39] he argues that such an eventuality would see a resumption of small-scale, decentralized, efficient and ecologically benign patterns of development ('eutechnic') that were stymied in the late nineteenth century, primarily by government subsidies to large corporations for transport and communications infrastructures that gave an insuperable advantage to large-scale producers.

This technologically-driven potential for informalization and micro-production off the books, may dramatically reduce the unit transformity costs of complexity—which is to say, it may allow us to generate 21st century levels of technical complexity with an ecological footprint closer to early modern capitalism—possibly, maybe ... who knows. However, by the same token, this potential is more disruptive than any previous round of creative destruction. Imagine the impact of driverless vehicles on American society, putting millions of largely male breadwinners out of work, possibly in less than a decade—and then multiply it by an order of magnitude.

From the vantage point of the current discussion, one obvious casualty would be the social compact that has kept the class peace since 1945. It is for this reason that the impact and potential of such technology cannot be considered separate from the broader political economy of the state and society, the structure of mutual identifications and we-imaginaries, the strength and vitality of family, church institutions, religious affiliations, patterns of local reciprocation and mutual aid and so on—which is really to say, the nature of the appropriate corresponding survival unit. Since there is every indication that limits to growth and technological innovation will conspire to undermine the social democratic state and that we may well see a radical contraction in the market, the question arises as to how Livelihood might function as a third dimension of a more embedded form of economy and culture. Table 1 summarizes potential characteristics of a Livelihood-based political economy across various societal domains.

**Table 1.** Political Economy of a Livelihood Economy.

| Domain | Characteristics |
|---|---|
| Technology and production | Distributed innovation and disruptive Maker economies (Kish 2018) |
| Smaller welfare state | Lower cost stripped down safety net, or devolved entirely to communities |
| Subsidiarity | Much greater powers of self-management for communities and households |
| Global economy | Reversal of globalization: cancellation of trade-pacts, focus on raw materials (where necessary), much reduced trade in food |
| Nation-state | Greater powers and regulatory reach vis-à-vis global economy; Reversal of nationalization: Less power/reach vis-à-vis regions, communities, households |
| Compensating survival units | Re-emergence of extended family, clan structures, religion and community forms of association as well as guilds, friendly societies, voluntary fraternities—as primary sources of physical and economic security. |

## 10. The Politics of Livelihood

Taking a cue from both Polanyi [9] and the early Herman Daly [40], the notion of Livelihood developed here is construed as a foundational domain—necessary to reign in both State and Market, and to create a three-legged political economy for an alternative modernity. Only livelihood offers the possibility of moving toward a low energy/material throughput version of modernity. But at the same time, the politics of livelihood potentially destabilizes 'business as usual', left-right politics.

Firstly, the policy commitments and cherished values of the social democratic left are now inextricably tied up with the idea of a progressive public sector and an interventionist state. It is almost impossible for most activists and opinion formers on the left to countenance an involuntary, let alone deliberate, reduction of the scale and scope of state activities. The prospect of devolving aspects of the social compact to communities, religious groups or families seems positively reactionary—and in many ways, it is. Conversely, separate from whether such a scenario is inevitable, there are some ways, even from a left-perspective, in which greater freedom for individuals, families and communities may

be a welcome prospect. At the same time, such a trajectory for ecological economics and politics has the potential to create opportunities for new alliances.

Since the 19th century, conservatism has always been riven by two very different visions of economy and society. The market conservatism associated with Thatcherism, the American 'neo-cons' or in Canada Stephen Harper, is associated with neo-liberalism for a reason. It is in many ways a branch of classic liberalism, emphasizing a small ('night watchman') state, a default commitment to laissez faire and a faith in the self-organizing power of the market. Many strands of social conservatism on the other hand—the Christian traditionalist Paleo-conservatism, the (pagan) French New Right of Alain de Benoist green communitarianism—place fidelity to community, national-civic obligation, family and a strongly developed notion of virtue, above a commitment to abstractions of the market or any ideological commitment to capitalism. Many traditionalists (not least the French New Right) are avowedly anti-capitalist and as hostile to the power of large corporations and free-wheeling global markets as they are to the over-weaning state.

All of this suggests at least some common ground with a variety of socially conservative and libertarian strands on the right that are worth exploring in advance of any serious shock to the economic system. At the same time, this analysis suggests also the necessity of difficult conversations with many more natural historic allies on the left, with regard to the sustainability of the interventionist state, the mutually constitutive relation between the state and the market and the ontology of unrestrained individualism that underlies the Enlightenment vision of human rights stripped of social and contextual obligations on the one hand, and consumer capitalism on the other.

## 11. A Pre-Figurative Example: Mental Health Care in the Community in Geel

In Belgium, there is a small town called Geel where for 700 years, families have fostered people with serious mental illnesses, taking them into their homes and treating them as family. Agreeing to become a foster family often means making a lifetime commitment to a stranger's care. It means accepting that person as family and taking primary responsibility for their wellbeing. For centuries, this practice has persisted with only minimal oversight and support by the state. Today, hospital-based medical care is available to manage pharmaceutical regimens, to deal with crises, and to ensure that people are being treated with dignity. But aside from this, almost all direct care is provided by families, in the home and in community settings, with the philosophy that full integration into community life is its own kind of therapy [41,42].

The family care system in Geel is a non-medicalized approach to mental illness. Until the past few decades, most of the time families did not even know the formal medical diagnoses of their boarders. Now, people are more likely to be aware of their boarder's diagnosis, but the overwhelming emphasis is still on finding ways to live together that work both for the boarder and their host family. Boarders are treated as "special" or "different", but never as "patients", the "mentally ill", or people who need "psychiatric" help [41].

Geel's practice of family care is rooted in Catholic traditions and dates back to the 14th century, when pilgrims began traveling to Geel to visit the shrine of St. Dymphna. The story goes that St. Dymphna was an Irish princess in the 7th century, the daughter of a Pagan king and a Christian queen. When Dymphna's mother died, her father went insane from the grief and commanded that Dymphna marry him. She ran away, but he caught up to her in Geel, where he beheaded her. A shrine was created in her honour, and she became known as a Saint with the powers to intervene on behalf of people experiencing mental illness. The shrine became a popular pilgrimage site beginning in the 1300s, and in the 1400s a hospital was built beside the church to accommodate the influx of visitors seeking help from St. Dymphna. It was soon too small for the number of people it attracted, and community members in Geel began accepting pilgrims into their homes [41].

Something like Geel is a world away from most community-based approaches to mental illness that exist in modern healthcare systems. Geel's model is uniquely successful at integrating people as full participants in their families and community. Its long history and deep entrenchment within a

specific place have made it resilient to the forces of modernity that would usually erode such practices. In many ways it is a hold-over from a pre-modern era when health and care activities took place mostly in the realm of reciprocity, the sphere of informal gifting between people who shared bonds of social obligation. In Geel, the family care system began because it was mutually beneficial [41]. People were motivated by Christian values of faith and charity to take people in, and in return for care and lodging, boarders would often work on the family farm or at local businesses.

Geel exists on the radical fridges of community-based care. While the families that participate receive a small stipend from the state, their motivation for participating in the practice is decidedly non-economic. It can also be argued that their motivation is non-rational. People do not choose to be host families in Geel because they have done a cost-benefit analysis and decided that it is worth the time and effort. Instead, they have a strong conviction that it is the right and moral thing to do, and a sense that caring for someone in this way gives their lives meaning. Accepting someone with serious mental illness into your own home, to live with you and your spouse, or perhaps with your children, is as you can imagine, not always easy. Take for example Toni Smit, who has taken in boarders for decades. During this time, her and her husband have learned to navigate many difficult behaviors. For one man, they had to repeatedly chase away the lions he saw coming out of his bedroom wall at night. For another, they had to deflect his overpowering physical attachment and affection for Smit, which had begun to put a strain on her marriage. Another host describes living with a man who would twist the buttons off of his shirt every day. At night, his host family would sew the buttons back on, and the next day he would twist them off again. In Geel, these behaviors are not treated as burdensome, but are just a part of life. These things are "quirks" rather than "problems", and people deal with them by finding social solutions, chasing the lions away, sewing on the buttons, rather than turning to pharmaceutical or other medical interventions [43].

Geel has clear implications for how we might structure health systems in a scenario of rapidly declining economic growth and mounting ecological crises. For instance, it gives us a sense of how we might transition health and care practices into a rejuvenated Livelihood sphere of social reciprocity that will need to expand to compensate for contracting Market and State sectors. But in doing so, it also has implications for the structure of the broader political economy in an alternative, more place-bound, low-energy modernity.

Practitioners and researchers who are working in the area of degrowth suggest that the shrinking economies of the future will be economies of health and care [44]. Policies such as a universal basic income and work-sharing could free up people's time to focus on their own and each other's health by growing food, cooking together, looking after children, and caring for aging parents. Lower incomes and fewer work hours will mean that people are more involved in making things for themselves, repairing their own homes, and participating in community life. Public health measures that emphasize population health will be prioritized over the expansion of pharmaceutical cures. A slower pace of life will result in lower rates of stress and anxiety disorders and declining availability of highly processed foods will reduce much of the burden of diet-related disease that we see today. In many ways, this appears to be a desirable future, though not without its own wicked tensions, for instance, related to the potential resurgence of traditional gender roles and ethnocentrism [6].

The central task for us as researchers is to understand the kinds of cultural commitments, values, rituals, and social arrangements that uphold niche health practices such as those in Geel. Geel cannot be transplanted. The family care system is rooted in place, in a specific cultural worldview, and in a unique history. The model is not necessarily amenable to 'scaling' in the way that you might scale a successful program-based approach to mental health. However, if we can understand why the practice has been so resilient, then perhaps we can begin to cultivate similar values and behaviors in our own local contexts. Geel is not the only practice that has this kind of potential. Others include:

- Care farming: A green approach to health that involves engaging people with mental health issues or other social vulnerabilities in meaningful agricultural activities that enhance physical, mental, and social wellbeing and that can restore local ecosystems [45,46].

- Intergenerational care for elders: Cohousing for seniors and other populations such as university students (e.g., Humanitas in the Netherlands), addressing the challenge of an aging population while strengthening community bonds of mutual obligation across generations [47].
- Txoko: Cooking clubs in the Basque region of Spain that emerged during the process of industrial modernization to give quasi-familial support to young men far from home. Organized around a kitchen house, these clubs involve rites of passage and life-long commitments to other members—and more often than not become a central institution of family life for the young men, their wives and children [48].

## 12. The Politics of Transition

As a perceptive reviewer commented, the danger of a thought experiment of this kind is that it perhaps underplays the problem of transition in favor of an ideal-type future. The Livelihood economy that we have intimated would not necessarily be very progressive [2]. With any contraction of the state, there would certainly be many losers. However, such a contraction must be a defining feature of any ecological–economic understanding of biophysical limits. The question then arises, what might it look like and are there any opportunities for more rather than less benign futures? Elsewhere [49] we have explored the potential for a libertarian-green agenda in rural Canada asking what it would take to get petrol-heads, Mennonites and red-neck truck drivers to vote for the Green. Thought experiments of this kind are revealing in surfacing radical policies that may serve simultaneously green Livelihood, social conservative and libertarian agendas. Such options might include things such as radical subsidiarity and deregulation in areas such as alcohol, meat, dairy production; home/free-schooling; vehicle licensing and insurance; planning land-use zoning; health and safety rules; insurance requirements; (weekly, annual, life-long, one off) national and community level conscription as a form of in-kind insurance contribution; or a minimal basic income tied to community participation, e.g., [50]. Our rationale for exploring this agenda has been that the transition models (such as they are) of either (a) eco-modernization e.g., [51,52] or (b) radical 'degrowth' e.g., [53] —will both fail in their own terms, unless they involve a politics that can gain traction with board swathes of society, including rural and conservative demographics. The default strategy seems to involve waiting for opportunities opened by some kind of corporate apocalypse and the collapse of global capitalism [54]. This partly explains why, at this moment of existential insecurity, the left in America and Europe has become so preoccupied with fighting the culture war. A more productive strategy would be to focus on creating a wedge between low- and middle-income conservative voters and taken-for-granted support for corporate capitalism. One way to do this would be to pit small-scale market capitalism against the corporations. Radical libertarian policies focusing on entrepreneurial freedom of action and community self-organization at the level of family, household and neighbourhood, could simultaneously legitimate higher-level corporate regulation and the partial re-nationalization and re-localization of economic life. This would involve inverting the current pattern whereby the unit costs of regulation and tax rise rapidly the smaller the scale and scope of the operation. Taking seriously the possibility of reducing global trade (particularly in food), re-emphasising the civil society of nation-states as the locus of political and economic control and forcing up-scale corporate producers to bear more fully the environmental costs of their business models—all of this clearly involves the partial reversal of economic globalization. But by the same token, it would require a step back from the cosmopolitan shibboleths of cultural globalism and the reassertion of a civil society rooted in imagined communities of citizens. This is a difficult political prescription because it involves taking seriously and embracing if at the same time re-imagining some of the intuitions and commitments that have animated populist and nationalist politics over the last three years. Nevertheless, if the intuitions of the degrowth camp are right, this kind of relocalization is anyway the political-economic landscape of the future. On the other hand, if the more optimistic diagnoses of the eco-modernists are correct, greens still need a populist breakthrough to engender the political traction that would allow rapid and directive regulation. From a political perspective, the key is to develop a suite of policies that differentiate varieties of capitalism—and

particularly the dynamics of place-bound markets embedded in the relational and normative matrices of communities, on the one hand, from the price-setting markets involved in global trade. It is worth noting that this understanding of Livelihood politics, including the commitment to make corporations as well as the state subordinate to communities and families, resonates strongly not only with left-libertarian political economy [38] but with influential strands of Burkean conservatism [55] as well as the anti-capitalism of the social-catholic theories of subsidiarity and 'distributism' [56].

## 13. Conclusions: Future Research

From a political economy point of view, the often under-acknowledged central dynamic of capitalist modernization has been the elaboration of the institutions and co-development of Market and State and the atrophy of the institutions of Livelihood based on affiliations of family, kindred, place, and tribe. This was a necessary process because all of the institutions of modern society that we cherish depend on individual citizenship, an institution that requires individuals to be detached from their local and familial survival units and to come to depend on the market and the state. Left-right politics tends to focus on the state vs. the market, but as Polanyi showed clearly, the state and the market are two sides of the same coin. Through the process of modernization, including the advent of universal literacy, the individual citizen that emerges is psychologically very different to the person who existed in small-scale societies. Today, regardless of their political affiliations, most conservatives, liberals and social democrats alike take these features of modern society for granted. And in fact, the modern personality is so taken for granted, that it is effectively invisible. Since the late-20th century, however, it has become apparent that these features of modernity come with an unbearable ecological price tag. Over the last 40–50 years, arguments about hard and soft sustainability, ecological modernization vs. limits to growth, and most recently the green new deal, hinge on this trade-off between ecology and social complexity. Conventional or mainstream sustainability perspectives often effectively ignore any zero-sum biophysical constraints to human action, at least in part because sustainability advocates often work in public policy arenas in which these ideas are highly unpalatable (see [57,58] for further discussion of mainstream sustainability perspectives). In contrast, degrowth perspectives, which do acknowledge the existence of zero-sum constraints to growth, do not realistically concede the relationship between energy and material throughout and taken-for-granted aspects of social complexity in modern societies. In the language of HT Odum [37], even seemingly intangible features of modern society such as the concept of human rights or equality before the law have high 'transformity costs.' Delivering these same societal features at significantly lower unit energy and material throughput is challenging and even conceivably impossible. Advocates of ecological modernization suggest that maintaining these desired features will require embracing technologies such as nuclear energy [51]. Degrowth perspectives reject the high-tech vision of ecological modernization but will not depart from the mantra that small is beautiful [53]. However, small is not always beautiful; historically, small has meant low tech and low freedom [2].

A fundamental feature of ecological economics as opposed to mainstream sustainability positions is that it recognizes biophysical limits to economic growth [40,59]. In light of this, from a policy and political point of view, the question of scale precedes configurations of social justice, which in turn should trump market efficiency. We have argued above that the trade-off between ecology and social complexity can be construed as a tension between the domain of Livelihood and the linked domains of State–Market. To preserve anything of the complexity of the State–Market, liberals and social democrats in an era of ecological constraints will need to find some accommodation with the domain of Livelihood. Within this adjacent possible, there may be ways of reorganizing modern society that allow humanity to step back from (or in some contexts adapt to) ecological crises as well as societal collapse. We might even find a path that is in some respects more socially benign and happier for more people in more countries than the one we are on at present. But we certainly cannot find this path unless we are willing to look for insights and inspiration across the whole breadth of the ideological landscape. This will require a much more nuanced and well-intentioned, honest conversation with

conservatives and libertarians. To the extent that ecological economics engages in this conversation, the adjacent possible ecological political economy grounded in a triptych survival unit of Livelihood, State and Market will come into clearer focus (see Sections 9–12).

Based on our analysis of the emergence of the State–Market survival unit during capitalist modernization, we argue that the parameters of an ecological political economy are defined by the selective recovery of the domain of Livelihood alongside the selective downsizing of the linked domain of the State–Market. The imperative of ecological scale requires a rebalancing of these domains in an alternative modernity; Livelihood must go up, while State–Market go down. At the same time, maintaining a liberal polity requires a minimum degree of complexity and individual mobility (see [2]). When scaling down the State–Market, it is essential to be selective. Few people who have become accustomed to such things likely want to live without a motorway system, the internet, universities, or modern dentistry. However, these elements have a high transformity cost that must be addressed in an ecological political economy. If it is feasible to reduce the unit transformity costs of certain functions, then it is possible to make small more beautiful as well as more complex. For example, technologies such as 3D printing open up space for high-tech localization that can deliver a technical product while stripping out ecologically costly levels of complexity such as packaging, transportation, etc. [60].

Any serious consideration of a Livelihood-based survival unit and political economy also needs a much firmer ontological and epistemological grounding. This is because the concept and trajectory depart so severely from the taken-for-granted tropes of tandem material and moral progress that Western society inherited from the Enlightenment. These questions have received little research attention from an ecological economics lens and could form the basis of future doctoral work in the discipline. A recurring feature of political philosophers who take biophysical limits seriously is a return to Aristotelian and Thomist philosophical frameworks. Thus, for example Ophuls's [61,62] groundbreaking and sobering engagement with the topic frames the discussion in terms of Aristotelian virtue ethics and the need to move away from deeply entrenched Enlightenment tropes of moral individualism, universalism and linear progressivism. Ophuls's work anticipated a resurgence in virtue ethics in the wake of Macintyre's After Virtue and more recently John Millbank's theological project of radical orthodoxy represents a move in the same direction and has achieved significant traction outside of academic theology, not least with Blue-Tory thinkers such as Philipp Bond associated with the ResPublica thinktank in the UK [63]. Macintyre's ideas have been taken most seriously by Catholic thinkers in America such as Rod Dreher whose The Benedict Option [56] advocates Christian communities withdrawing and disengaging from the ecologically devastating and morally nihilistic reality of consumer capitalism. In this context, priority research areas for ecological economics would include:

(1) Livelihood as a basis for radical political economy in traditions such as (Social Catholic) Distributism and Guild Socialism;
(2) The philosophical basis for a green Livelihood politics in Aristotelian virtue politics and the work of Alasdair MacIntyre;
(3) The potential for economic and societal disruption arising from new fabrication technologies and their likely impact on formal structures of the market and the fiscal basis of the state
(4) Unexpected and perhaps paradoxical alignments between the values and practices of small-c conservatives and social-democratically minded greens, especially related to Livelihood activities including child-care, elder-care, DIY/informal economic activity, household organization;
(5) A survey of Livelihood-related innovations and the historical repertoire of self-organized, familial and community forms of support (such as the Txoko) from different societies and periods.

**Author Contributions:** Conceptualization, S.Q., K.Z. Writing—Original Draft Preparation, S.Q., K.Z.; Writing—Review and Editing, S.Q., K.Z. Visualization, S.Q.; Supervision, S.Q.

**Funding:** This research received no external funding.

**Acknowledgments:** Katharine Zywert's doctoral research is funded through the Ontario Graduate Scholarship and the President's Graduate Scholarship, University of Waterloo.

**Conflicts of Interest:** The authors declare no conflict of interest.

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
