# Peer review of "Livelihood, Market and State: What does A Political Economy Predicated on the ‘Individual-in-Group-in-PLACE’ Actually Look Like?"

_sustainability, doi:10.3390/su11154082_

Round 1

Reviewer 1 Report

Authors have improved their manuscript significantly comparing to the previous version. Currently, the main aim of the paper, methodology and conclusions are clarified, what makes the article more readable and consistent.

Author Response

Thank you for your positive comments on the revised version. As suggested, we have carefully edited the paper to improve grammatical/typographical issues. 

Reviewer 2 Report

Authors have made an effort to improve their paper and respond to my concerns. However, there are some issues that still need to be addressed more effectively:

- to what extent does this paper differ from previous papers published by authors? Please, emphasize that. In the Introduction section (page 2, line 57) authors claim "...the paper then attempts to discern the parameters and constraints on an adjacent possible ecological political economy…". Both the terms "parameters" and "constraints" are not referred to in the analysis and conclusion section.

- the sustainability issue should be strengthened, taking into account the journal readership.

Author Response

Thank you for taking the time to review the second draft. Based on your feedback, we have made the following changes to the paper: 

Comment:

- to what extent does this paper differ from previous papers published by authors? Please, emphasize that. 

Response:

We have added text to the introduction to more precisely delineate how this paper differs from previously published work.

Comment:

- In the Introduction section (page 2, line 57) authors claim "...the paper then attempts to discern the parameters and constraints on an adjacent possible ecological political economy…". Both the terms "parameters" and "constraints" are not referred to in the analysis and conclusion section.

Response: 

We have revised the conclusion to directly highlight the parameters of an ecological political economy. We also made it clear in section 9 that Figure 3 illustrates the parameters of a triptych Livelihood/State/Market survival unit. We have removed the reference to "constraints" in the introduction.  

Comment: 

- the sustainability issue should be strengthened, taking into account the journal readership.

Response:

We have revised both the introduction and conclusion to emphasize how the central argument of the paper is relevant to sustainability studies. 

Reviewer 3 Report

see attachment

Author Response

Thank you for taking the time to review the second draft of our paper. Based on your feedback, we have made the following changes: 

Comment:

 The paper focus on ecological economics should be more clearly explained.

Response: 

In the introduction and conclusion, we have expanded on the relevance of the paper’s central argument to ecological economics. 

Comment: 

ENGLISH: The paper’s English is sufficient, however, there are a number of grammatical errors which should be carefully revised – mostly typographical mistakes that a solid English editor (or native) could fix with a read through. 

Response: 

We have carefully edited the paper to correct grammatical and typographical mistakes. 

Comment: 

Conclusion: a proper conclusion should be written. It should be (sufficiently) lengthened by wrapping together all of the points of the paper. I was expecting a strong finish, instead the paper ends abruptly with a few bullet points. 

Response: 

We have expanded the conclusion to 1) review the arguments put forward in the paper, 2) emphasize how these arguments make a novel contribution to both ecological economics and sustainability, and 3) outline the parameters of a potential ecological political economy. 

Comment: 

Figure 1, 2, and 3 should be redone, with a higher resolution, they are of poor visual qualiy 

Response: 

 Higher resolution images have been provided to improve the visual quality of the figures.

Comment:  

The sections of the manuscript regarding the information headings: "Author Contributions", "Funding", and "Acknowledgments" have not been inserted in the manuscript. Please use the recommended journal template structure for MDPI Sustainability. 

Response: 

We have added the following relevant sections to conform to the required structure for MDPI sustainability: “acknowledgements”, “Author Contributions”, “Conflicts of Interest.” 

Round 2

Reviewer 2 Report

Authors have largely addressed my concerns.